# Long- and Short-Term Glucosphingosine (lyso-Gb1) Dynamics in Gaucher Patients Undergoing Enzyme Replacement Therapy

**DOI:** 10.3390/biom14070842

**Published:** 2024-07-12

**Authors:** Pawel Dubiela, Paulina Szymanska-Rozek, Piotr Hasinski, Patryk Lipinski, Grazina Kleinotiene, Dorota Giersz, Anna Tylki-Szymanska

**Affiliations:** 1Department of Regenerative Medicine and Immune Regulation, Medical University of Bialystok, 15-089 Bialystok, Poland; paweldubiela89@gmail.com (P.D.); dorota.giersz@me.com (D.G.); 2Department of Pathophysiology and Allergy Research, Medical University of Vienna, 1090 Vienna, Austria; 3Faculty of Mathematics, Informatics and Mechanics, University of Warsaw, 00-927 Warsaw, Poland; p.szymanska@gmail.com; 4Department of Internal Medicine and Gastroenterology, Municipal Hospital, 43-100 Tychy, Poland; piotr_hasinski@wp.pl; 5Institute of Clinical Sciences, Maria Skłodowska-Curie Medical Academy, 00-136 Warsaw, Poland; patryk.lipinski.92@gmail.com; 6Faculty of Medicine, Vilnius University, 03101 Vilnius, Lithuania; grazina.kleinotiene@santa.lt; 7Department of Pediatrics, Nutrition and Metabolic Diseases, The Children’s Memorial Health Institute, 04-736 Warsaw, Poland

**Keywords:** biomarker, enzyme replacement therapy, Gaucher disease, glucosylsphingosine, lyso-Gb1

## Abstract

**Background**: Gaucher disease (GD) is a lysosomal storage disorder caused by mutations in the *GBA1* gene, leading to β-glucocerebrosidase deficiency and glucosylceramide accumulation. **Methods**: We analyzed short- and long-term dynamics of lyso-glucosylceramide (lyso-Gb1) in a large cohort of GD patients undergoing enzyme replacement therapy (ERT). **Results**: Eight-years analysis of lyso-Gb1 revealed statistically insignificant variability in the biomarker across the years and relatively high individual variability in patients’ results. GD type 1 (GD1) patients exhibited higher variability compared to GD type 3 (GD3) patients (coefficients of variation: 34% and 23%, respectively; *p*-value = 0.0003). We also investigated the short-term response of the biomarker to enzyme replacement therapy (ERT), measuring lyso-Gb1 right before and 30 min after treatment administration. We tested 20 GD patients (16 GD1, 4 GD3) and observed a rapid and significant reduction in lyso-Gb1 levels (average decrease of 17%; *p*-value < 0.0001). This immediate response reaffirms the efficacy of ERT in reducing substrate accumulation in GD patients but, on the other hand, suggests the biomarker’s instability between the infusions. **Conclusions**: These findings underscore lyso-Gb1’s potential as a reliable biomarker for monitoring efficacy of treatment. However, individual variability and dry blood spot (DBS) testing limitations urge a further refinement in clinical application. Our study contributes valuable insights into GD patient management, emphasizing the evolving role of biomarkers in personalized medicine.

## 1. Introduction

Gaucher disease (GD) (OMIM^®^: 230800) is an autosomal recessive lysosomal storage disease caused by biallelic pathogenic variants in the *GBA1* gene, resulting in a deficiency in β-glucocerebrosidase (GCase), impairing the metabolism of ceramides, and leading to a build-up of glucosylceramide (Gb1) in lysosomes [1]. The accumulation of Gb1 mostly affects macrophages, changing their structure, enhancing their proliferation, and infiltrating into the bone marrow, spleen, and liver [2]. The disease manifests with thrombocytopenia, anemia, and pancytopenia, along with an enlarged spleen and liver and bone involvement (GD type 1, GD1). Some *GBA1* variants are accompanied by central nervous system (CNS) involvement (GD types 2 and 3, GD2 and GD3) [3,4]. A clear phenotype–genotype correlation for common *GBA1* variants can be observed, including the c.1226A>G (p.Asn409Ser) [N370S] variant as the most common in Caucasians, which protects from CNS involvement [5,6], while homozygosity for the c.1448T>C (p.Leu483Pro) [L444P] variant determines the neuronopathic GD type 3 phenotype [7,8].

The gold standard in the diagnosis of GD was (and still is) a method based on measuring GCase activity in peripheral blood cells, followed by *GBA1* next-generation sequencing. Recently, a dried blood spot (DBS) test is in common use, utilizing the same diagnostic protocol with the addition of biomarkers [9,10,11,12,13]. The first described and studied biomarker was chitotriosidase (ChT), massively produced by the activated macrophages [14]. Chitotriosidase is still a reliable biomarker, quantitatively reflecting the disease’s initial severity, its progress, and the effectiveness of treatment. The method of collecting and assessing chitotriosidase activity is relatively simple and has remained the same for over 20 years, which guarantees comparability of results between years. However, it was found to be limited by the common presence (5% of the population) of null *CHIT1* allele homozygotes and hypomorphic alleles (p.G102S) that are also reported to affect chitotriosidase activity [14,15,16,17].

In 2010, a newly introduced biomarker was expected to provide a breakthrough in the lysosomal storage disorders field. An alternative metabolic pathway favored in states of GCase deficiency has been identified, where loads of Gb1 undergo deacylation due to acid ceramidase activity, producing glucosylsphingosine (also named lyso-glucosylceramide, lyso-Gb1) [17].

Lyso-Gb1 also distinctly reflects the disease progression and treatment effectiveness. Rolfs et al. have observed a significant reduction in lyso-Gb1 after enzyme replacement therapy (ERT) initiation, from a median of 200 ng/mL before the start of treatment to levels below 50 ng/mL after therapy onset and the immediate elevation of the biomarker when the therapy was ceased [18].

Since GD is chronic by its nature and treatment success is influenced by many factors, i.e., a pathogenic *GBA1* variant, age at time of appearance of symptoms and diagnosis, and the dose of ERT, we aimed to evaluate lyso-Gb1 in a long-term observational study (up to 8 years) in the same cohort of ERT-treated patients with GD. In addition, we performed a pharmacokinetic study on lyso-Gb1 on the subgroup of samples measuring levels of the biomarker before and right after enzyme infusion.

## 2. Materials and Methods

### 2.1. Patients

In total, 75 GD patients (56 GD1 and 19 GD3) treated with ERT between 2016 and 2023 were recruited for the study. The blood samples were obtained at one time point each year (the end of November/beginning of December) from GD patients undergoing annual clinical follow-up. The clinical assessment was standardized and performed by a single experienced clinician. However, not all the patients had measurements taken every year. For the sake of proper methodology, in the long-term analysis, we considered only those patients who had at least six measurements noted, i.e., 32 GD type 1 patients and 15 GD type 3 patients (Appendix A). The collection of DBSs for lyso-Gb1 assessment followed the scheduled visits and could occur at any time point within the bi-weekly treatment regimen.

The second part of the study was the analysis of the short time response of lyso-Gb1 to ERT. Lyso-Gb1 measurement on the day of infusion was performed on 26 blood samples from 20 GD (16 GD1 and 4 GD3) patients. The experiment was duplicated for 2 patients and performed in triplicate for 2 patients. This study was performed as follows: the DBS samples were taken from patients coming routinely for every-other-week ERT infusions at two time points, right before the start of the infusion and 30 min after the end of the infusion.

All GD study patients were treated with ERT with imiglucerase (Cerezyme^®^; Genzyme/Sanofi) or velaglucerase alfa (VPRIV^®^; Shire/Takeda) for a minimum of 15 years and a maximum of 28 years. The dose of ERT was adjusted based on disease type and clinical picture: 30 U/kg/every other week (EOW) for type 1 GD and 60 U/kg/EOW for type 3 GD.

The protocol of the study was approved by the local Ethical Committee of The Children’s Memorial Health Institute, Warsaw, Poland (number 51/KB/2019). Written informed consent was obtained from all the participants. The study was conducted in accordance with the ethical principles outlined in the Declaration of Helsinki.

### 2.2. Sample Processing and DBS Analysis

DBS tests were performed as suggested by the test’s producer. 

Archimed life is a laboratory certified with ISO 15189 (Medical Laboratory*—Clinical Chemistry to Genetics), ISO 9000 (Quality Management System), ISO 13485 (Medical Devices—IVD Development and Production), and GLP-lab certificates and fully integrated for clinical studies [13,19]. According to the laboratory information, the sample is analyzed as previously described [11]. Lyso-Gb1 is tested directly from the DBS card with specific determination and quantification by multiple reaction monitoring mass spectrometry. Once the sample is positive, the *GBA1* gene is sequenced with Sanger and next-generation sequencing platforms [19].

### 2.3. Statistical and Quantitative Analysis

The purpose of the statistical analysis was to investigate to what extent lyso-Gb1 is a stable and, therefore, reliable biomarker to assess GD severity and evolution, as well as therapy efficacy. “Stability”, from the statistical point of view, means that the averages for each year, calculated for the subcohort of patients who had all the measurements taken, should be comparable. We thus performed a comparison of the means of lyso-Gb1 with a paired *t*-test for this group of patients.

Stability can also be looked at as a relatively small variance calculated for each patient for the whole time interval of observations (8 years). To assess this, we could simply calculate the variance for each patient, but since the levels of lyso-Gb1 varied significantly between them, we needed a unitless quantity independent of the mean, i.e., the coefficient of variation (standard deviation divided by the mean).

The analysis of the short time response of the lyso-Gb1 level to ERT infusion was performed with a paired *t*-test for the 20 patients for whom we measured the biomarker level right before and right after the infusion. For those patients, who had two or three measurements taken, we first calculated the mean value so that every patient had a single pair of results, and thus the average for the whole group was not distorted by repeated results.

## 3. Results

### 3.1. Patients’ Characteristics and Long-Term Analysis

Our dataset comprised 75 GD (56 with GD1 and 19 with GD3) patients undergoing ERT, all receiving stable and optimal doses of treatment (Appendix A and Appendix A). As mentioned before, for specific statistical testing or proper data visualization, we did not take into account patients who had too few results. For example, the visualization for the 8-year observation was performed for patients who had six or more results available, i.e., 32 GD type 1 patients and 15 GD type 3 patients (Appendix A). The raw results for these patients are presented in Figure 1, and since the scale of the biomarker level differs among patients, we also depicted it in Figure 2, where the results are normalized to the first measurement; i.e., all were divided by the lyso-Gb1 level measured in a given patient in the year 2016.

Looking at Figure 2 with one’s bare eyes only, one could obtain the impression that the results vary substantially year to year and for each patient separately. In order to check the validity and statistical significance of this observation, we selected a subgroup of 15 GD type 1 patients and 9 GD type 3 patients who had a complete set of eight measurements taken. In this group, we performed a paired *t*-test of the lyso-Gb1 results, cross-comparing each year with other years. The *p*-values were all drastically above 0.05, except for four pairs of years. However, even the smallest of them (0.003), after a correction for multiple testing (24 comparisons performed), turned out to be insignificant. We thus restrained ourselves from claiming that throughout the observed period there was a change in laboratory method or some kind of mistake in probe treatment. Box plots are presented for a restricted range so that the inclusion of upper outliers (Q3 + 1.5 × IQR) does not distort the scale (see Figure 3).

We also verified the impression that GD3 patients exhibit less variability across years than GD1 patients. The unitless and mean-independent (which differs substantially among patients) quantity, that is, the coefficient of variation, was 34% on average for GD1 patients and 23% for GD3 patients. An unpaired *t*-test for the coefficient of variation for patients with at least six measurements resulted in a *p*-value of 0.0003 (see Figure 4).

### 3.2. Short-Term Observation

The cohort comprised 16 GD1 and 4 GD3 patients. Since the experiment was duplicated for two patients and performed in triplicate for two patients (Table 1), we first averaged the results for these four patients and then calculated a paired *t*-test, obtaining a *p*-value of 0.008. The comparison between the values before and right after ERT administration, normalized to the result recorded before ERT administration, is presented in Figure 5. The average reduction in the lyso-Gb1 level was by 17% and a paired *t*-test conducted for the normalized data resulted in a *p*-value of less than 10^−4^; we therefore claim that the decrease in lyso-Gb1 level just after the infusion of ERT cannot be a result of randomness.

## 4. Discussion

Lyso-Gb1 has emerged as an important biomarker in the management of GD, providing a valuable insight into disease progression and treatment response. Previous studies showed that lyso-Gb1 can be a good indicator of GCase deficiency through substrate accumulation, demonstrating an association with disease severity, reflecting the imbalance in sphingolipid metabolism [20]. Thus, lyso-Gb1 plays an essential role in monitoring treatment responses with ERT. Our data aim to delve into the intricate dynamics of lyso-Gb1 in the context of GD management. In addition to an eight-year observational study, our short-term analysis of immediate ERT effects on lyso-Gb1 concentrations provides a more granular understanding of the rapid response to treatment. Our investigation reveals also notable variability in lyso-Gb1 levels, both at the time of infusion and over the years.

Our longitudinal study revealed a noteworthy but not statistically significant temporal fluctuation in lyso-Gb1 levels over the eight-year observation period. Contrary to the results from previous studies, lyso-Gb1 concentrations did not consistently correlate with the severity of GD and treatment outcomes [20,21,22,23]. This temporal variation challenges the conventional notion of a linear relationship between lyso-Gb1 concentration and disease progression, suggesting substantial randomness involved in either the laboratory method or the intrinsic variable nature of the biomarker itself. Since the difference can be explained neither by dosage modification nor by changes in clinical picture, we investigated the short time scale of the biomarker dynamics as well.

This short-term observation of lyso-Gb1 levels right before and 30 min after ERT infusion revealed a rapid and significant reduction in lyso-Gb1 concentration. We observed an acute but still noisy reduction in lyso-Gb1 levels during this short-term analysis. This finding emphasizes the need for comprehensive investigations into the temporal dynamics of treatment-induced alterations in lysosomal pathways. It also raises the question of how stable the treatment response over the 2-week intervals between infusions is. There are no data available, and our study is the first of its kind.

While DBS testing has become a convenient and widely employed method for assessing lyso-Gb1 levels, it is crucial to acknowledge its limitations highlighted in the study. The weaknesses associated with DBS testing in GD can be triggered by many factors in the clinic but also in the laboratory. Sampling variability can be increased by variations in sample collection, storage, and processing or changes in hematocrit [24,25]. The variability could introduce difficulties in interpreting results accurately and lead to inappropriate dose escalation/reduction. Additionally, the process of applying blood onto filter paper can artifactually alter other analytes, potentially affecting the reliability of the DBS methodology and the interpretation of lyso-Gb1 levels as a widely accepted tool to monitor patients in daily practice [26].

## 5. Conclusions

In conclusion, our combined long-term and short-term analyses present a comprehensive overview of lyso-Gb1 dynamics in GD patients undergoing ERT. Our data shed light on the intricate dynamics of lyso-Gb1 in GD patient management, unraveling the complexities that extend beyond conventional paradigms.

The short-term observations affirm the rapid and tangible impact of ERT on lyso-Gb1 concentrations, providing valuable insights into the immediate treatment response. On the other hand, they raised the question of the stability of the treatment response during a biweekly interval.

Our data on lyso-Gb1 in GD patients revealed that the conspicuous “noise” in the biomarker level in our cohort across the 8 years of observation is not statistically significant. However, we checked that the individual variability independent of intrinsic individual biomarker dynamics is high, therefore acknowledging both the limitations of DBS testing and the urge to redefine the role of lyso-Gb1 in the evolving landscape of GD.

Further work should be performed to assess the value of lyso-Gb1 as a biomarker in the clinic and to elucidate the pathomechanism of Gb1 (glucosylceramide) and lyso-Gb1 (glucosphingosine, lyso-glucosylceramide) storage.

## Figures and Tables

**Figure 1 biomolecules-14-00842-f001:**
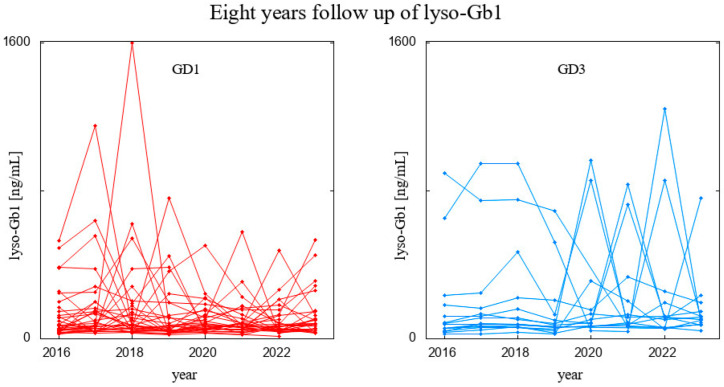
Eight-year follow-up on lyso-Gb1 levels in patients with GD1 (**left** panel) and GD3 (**right** panel), depicting raw data, with every trajectory corresponding to a patient.

**Figure 2 biomolecules-14-00842-f002:**
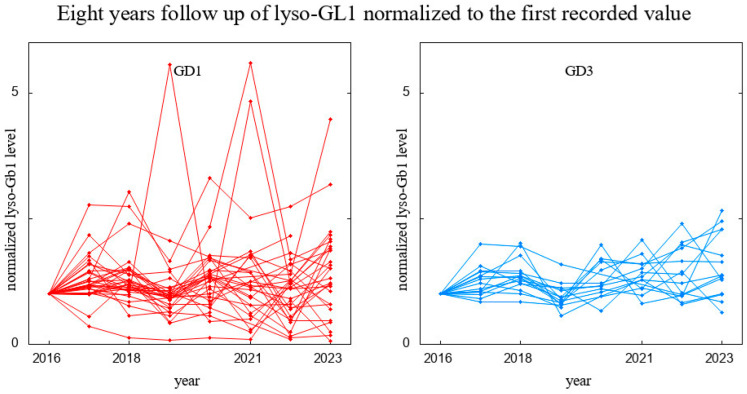
Eight-year follow-up on lyso-Gb1 levels in patients with GD1 (**left** panel) and GD3 (**right** panel). Data normalized to the first recorded value; i.e., for every patient, their lyso-Gb1 measurements were divided by the value recorded in 2016.

**Figure 3 biomolecules-14-00842-f003:**
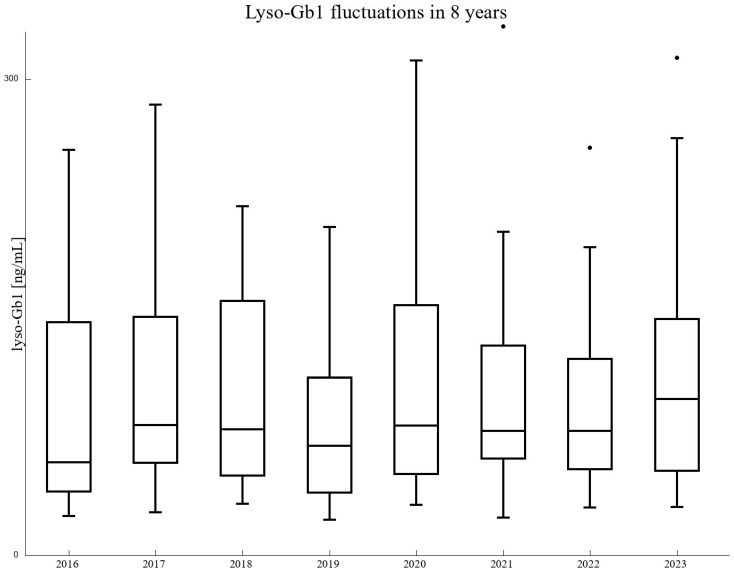
Box and whisker plots for lyso-Gb1 fluctuating during the 8 analyzed years. For the sake of clarity, the range was restricted to 300 ng/mL.

**Figure 4 biomolecules-14-00842-f004:**
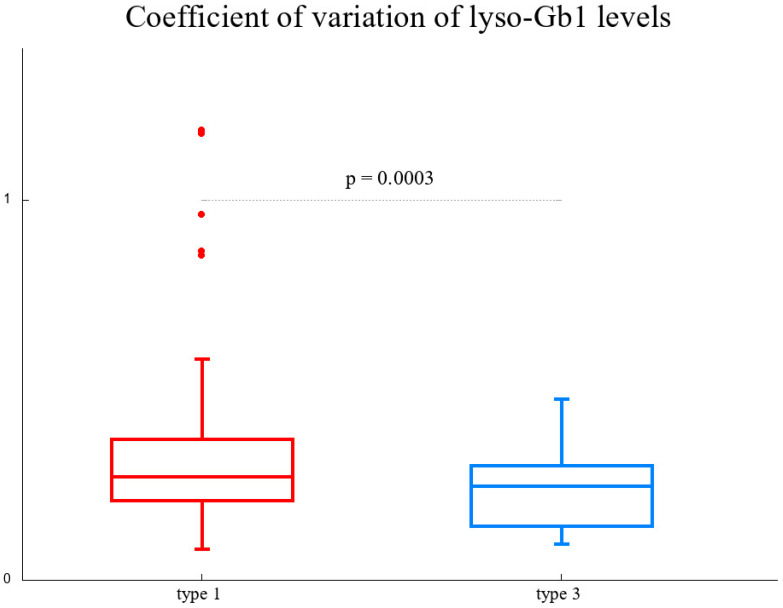
Box and whisker plots comparing the unitless value of the coefficients of variation (relative standard deviation, i.e., standard deviation divided by the mean) of lyso-Gb1 levels for GD1 (**left**, red) and GD3 (**right**, blue) and the corresponding *p*-value of an unpaired *t*-test.

**Figure 5 biomolecules-14-00842-f005:**
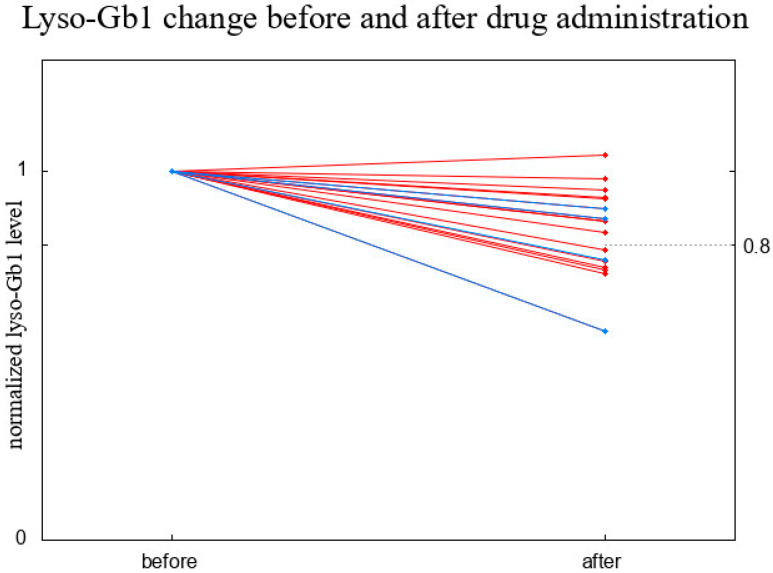
Short-term changes in lyso-Gb1 levels—a comparison between the value before and right after ERT administration, normalized to the result recorded before ERT administration. Red lines correspond to GD1 and blue lines to GD3.

**Table 1 biomolecules-14-00842-t001:** Short-term dynamics of lyso-Gb1 response to enzyme replacement therapy in GD patients.

Patient ID	GD1/3	Genotype	Lyso-Gb1 before [ng/mL]	Lyso-Gb1 after [ng/mL]
7	1	D438H/R87W	42	41.4
			31.2	34.9
20	1	N370S/c.1085C>T	50.6	47.9
28	1	D448G/R202X	54.3	51.2
			55	50
32	1	Unknown	45.7	39.8
35	1	N370S/L444P	33.3	24.6
NEW PATIENT ID1	1	N370S/N370S	63	49.4
NEW PATIENT ID2	1	N370S/L444P	47.6	42.7
NEW PATIENT ID3	1	N370S/ R202X	43.9	33.2
NEW PATIENT ID4	1	N370S/L444P	47.8	27.1
NEW PATIENT ID5	1	N370S/L444P	546	471
NEW PATIENT ID6	1	N370S/L444P	563	469
NEW PATIENT ID7	1	N370S/N370S	44.6	38.5
NEW PATIENT ID8	1	N370S/L444P	34.4	25.1
NEW PATIENT ID9	1	N370S/c.1085C>T	47.9	46.9
26	1	N370S/ R202X	42.4	30.6
NEW PATIENT ID10	1	N370S/c.1085C>T	37.3	34.6
NEW PATIENT ID11	3	L444P/L444P	52.3	39.6
59	3	L444P/L444P	59.7	33.7
71	3	L444P/L444P	145	148
			96.3	67.4
			71.4	65
72	3	L444P/L444P	53.9	48.5
			49.6	41.5

## Data Availability

More data are available upon request to the corresponding author.

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
