# Peer review of "Long- and Short-Term Glucosphingosine (lyso-Gb1) Dynamics in Gaucher Patients Undergoing Enzyme Replacement Therapy"

_biomolecules, 2024, doi:10.3390/biom14070842_

Round 1
Reviewer 1 Report
Comments and Suggestions for Authors
A General appreciation
- The subject of this article is very interesting.
In addition, the team has extensive experience of Gaucher disease and its pathophysiology.
- The article should be accepted but needs to be improved in terms of presentation (see line-by-line corrections). The authors should be more precise and explicit in all the tables and figures. Legends are too brief and not precise enough.to make figures and tables easy to understand.
- Regarding the design of the study, a flow chart (beginning with the 75 patients) should be drawn up to describe the inclusions for each type of sample taken for each group.
- I'm not an expert in mathematical methodology, but the authors or another reviewer more expert in the filed should be able to tell us whether it's possible to explain the results more simply.
- Regarding the discussion : i) what assumption do the authors put forward in favour of the technical bias hypothesis to explain the variations of the lyso-Gb1 results ? ii) what assumption do the authors put forward in favour of the hypothesis of the link of Lyso-Gb1 with severity of GD, treatment, time duration, dosage etc.
- There should be a table, even a brief one, on the clinical and biological characteristics of Gaucher disease, at least at the start of treatment.
B Specific comments made line by line
- Line 41-42
The accumulation of Gb1 mostly affects macrophages, changing their structure, enhancing their proliferation and infiltration into the bone marrow, spleen and liver. Is it possible to provide a reference for this particular sentence on the fact that Gb1 accumulation increases macrophage proliferation ?
- Line 49
L444P variant determining neuronopathic GD type 3 phenotype.
The new genotype nomenclature should be applied. It should be used in manuscripts by the entire community involved in Gaucher disease while I understand that the transition is difficult.
- Line 53
…tocol with addition of biomarkers that The first described and studied biomarker was chi-…
I do not understand this sentence.
- Line 61
….ue to an acid ceramidase activity, producing glucosylsphingosine (lyso-glucosylceramide, lyso-Gb1). For readers who are less familiar with Gaucher disease, can you clarify whether these 3 molecules: glucosylsphingosine, lyso-glucosylceramide, lyso-Gb1, are strictly identical in which case it is just the name that changes or whether there are minor differences ?
- Line 67
…..ness. Rolfs et al. has observed a significant reduction in -lyso-Gb1 after enzyme replace- There is probably a typo
- Line 68
….ment therapy (ERT) initiation: from median of 200ng/mL before treatment start to levels
A space is missing : 200 ng/mL
- Line 83
…….ical follow-up. The clinical interview was standardized and performed by a single experi- The term clinical interview is not used much in medicine. Is it just a telephone interview or a clinical examination ?
- Line 79 to 93 You should make a figure with a flowchart which would be easier to understand because it is not clear which patients have been chosen for which tests
- Line 94 to 96
All GD study patients were treated with ERT with imiglucerase (Cerezyme®; Genzyme Corporation, Cambridge, MA, USA), Genzyme) or velaglucerase alfa (VPRIV®; Shire Human Genetic Therapies, St Helier, Jersey)
The pharmaceutical companies Genzyme and Shire no longer exist, as these companies have been acquired by Sanofi and Takeda. You can put the old names because most patients were treates before the changing but in that case you must also put the new names. The other option is to only put the current names Sanofi and Takeda.
- Line 97 to 98
The dose of ERT was adjusted basing on disease type and clinical picture; 30 U/kg/every other week (EOW) for type 1 GD, and 60 U/kg/EOW for type 3 GD.
Is this the initial dose at the start of treatment or the dose at the time of sampling? Or is it the average dose ?
Were there any variations in the dose ?
- Line 122
…should be comparable. We thus performed a comparison of the means of lyso-Gl1 with a …..
Here you name lyso-Gb1 lyso-Gl1 , you need to homogenise
- Line 126
….variance for each patient, but since the levels of lyso-Gl1 varied significantly between Same remark as above : you named lyso-Gb1 lyso-Gl1 , you need to homogenise
- Line 145
.....urement, i.e., all were divided by lyso-Gl1 level measured in a given patient in year 2016. Same remark as above : you named lyso-Gb1 lyso-Gl1 , you need to homogenise
Line 174-175
Normalized data, i.e., such that the measurement before the is presented in Figure 5. The average
I do not understand this sentence
Line 179
Table 1. Short term observation. The name of this table is not explicit!! If readers do not have time to read the whole article, they should be able to find enough information in the table: name, etc. , legends must be detailed, the column ID it is not clear. The tabulation of the last row is non-aligned.The genotype must be written with the new nomenclature.
- Line 181
Figure 5. Dynamic Changes in Lyso-Gb1 Levels Pre- and Post-ERT Administration. This figure needs precision in the units and again homogenise lyso GL1 or Lyso GB1 in abscissa and ordonate
- Line 231
Further work should be done to assess the value of glucosphingosine as a biomarker in the clinic and to elucidate the pathomechanism of Gb1 and glucosphingosine storage.
In this sentence you finish as in the introduction with the molecule glusosphingosine and not Lyso-Gb1. The precise definition of each molecule need to be better defined (is it the same molecule or not) ?
Supplementary Table 2 Lyso-gb1 in years in GD1 and GD3 cohorts
|
|
Mean lyso-gb1 concentration [ng/ml/patient] |
|||||||
|
Year |
2016 |
2017 |
2018 |
2019 |
2020 |
2021 |
2022 |
2023 |
|
GD1 with all results (n = 16) |
110 |
159 |
178216 |
110 |
100 |
83 |
77 |
100 |
|
GD3 with all results (n = 9) |
190 |
83 |
216 |
155 |
202 |
196 |
210 |
199 |
The number in red is very high
Comments on the Quality of English Language
Minor revisions. Some sentences has to be checked
Author Response
Reviewer 1
A General appreciation
- The subject of this article is very interesting.
In addition, the team has extensive experience of Gaucher disease and its pathophysiology.
- The article should be accepted but needs to be improved in terms of presentation (see line-by-line corrections). The authors should be more precise and explicit in all the tables and figures. Legends are too brief and not precise enough.to make figures and tables easy to understand.
- Regarding the design of the study, a flow chart (beginning with the 75 patients) should be drawn up to describe the inclusions for each type of sample taken for each group.
We added following flow chart as the Supp. Fig. 1.
- I'm not an expert in mathematical methodology, but the authors or another reviewer more expert in the filed should be able to tell us whether it's possible to explain the results more simply.
Thank you for your comment. We appreciate your feedback and understand the importance of presenting the results in an understandable manner. While we have aimed to present the results in a clear and accessible way, we are open to more specific suggestions for improvement. We welcome input from experts in the field to ensure that our findings are communicated effectively. If you have any specific recommendations for simplifying the explanation of the results, we would be grateful for your insights.
- Regarding the discussion : i) what assumption do the authors put forward in favour of the technical bias hypothesis to explain the variations of the lyso-Gb1 results ? ii) what assumption do the authors put forward in favour of the hypothesis of the link of Lyso-Gb1 with severity of GD, treatment, time duration, dosage etc.
Thank you for your insightful comment. In our study, we have put forward certain assumptions to address the variations observed in the Lyso-Gb1 results and the potential associations with the nature of the momentum of biomarker measurement between ERT treatment cycles. We believe that these variations can be attributed to the following factors related to technical bias hypothesis and pharmaco-variability related to nature of the bi-weekly ERT cycles.
Every year, in order to assess the effects of therapy, we test biomarkers in our patients with GD. We have noticed that there are differences between subsequent test results. This made us feel a bit confused. This makes it difficult to assess the effectiveness of dose, therapy, etc. We collected eight years of results to assess the sensitivity and value of this biomarker, mainly for clinical work.
Our discussion assumes that the observed variations in Lyso-Gb1 results may be influenced by technical biases inherent in the methods of biomarker measurement. These technical biases could stem from factors such as sample handling, assay variability, or limitations in the methodology used for dried blood spot (DBS) assessments.
We acknowledge the complexity of interpreting biomarker dynamics and the need to consider various factors that may contribute to the observed variability in Lyso-Gb1 levels. The nature of momentum of biomarker measurement and the limitations of DBS assessment methodology are crucial aspects that we have considered in our study, and we appreciate your interest in these nuanced discussions.
- There should be a table, even a brief one, on the clinical and biological characteristics of Gaucher disease, at least at the start of treatment.
Thank you for your valuable comment. We appreciate your insight into the importance of providing a comprehensive understanding of the patient population under study. In our study, the design focused on patients undergoing long-term stable treatment, and as such, treatment-naive patients were not included in the cohort. Consequently, we do not have data pertaining to the clinical and biological characteristics of treatment naive Gaucher disease patients at the start of treatment.
While we acknowledge the potential value of such information, the nature of our study limits our ability to provide this specific dataset.
B Specific comments made line by line
- Line 41-42
The accumulation of Gb1 mostly affects macrophages, changing their structure, enhancing their proliferation and infiltration into the bone marrow, spleen and liver. Is it possible to provide a reference for this particular sentence on the fact that Gb1 accumulation increases macrophage proliferation ?
The statement regarding Gb1 accumulation and its potential impact on macrophage proliferation is supported by Pandey and Grabowski in their publication "Immunological cells and functions in Gaucher disease" (Crit Rev Oncog. 2013;18(3):197-220). The authors discuss the mechanisms by which Gb1 accumulation can influence macrophages, highlighting three main pathways:
- Lysosomal Stress: The excessive buildup of Gb1 causes lysosomal stress, leading to the release of pro-inflammatory cytokines and other signalling molecules.
- Inflammatory Response: Engorged macrophages secrete various cytokines (e.g., TNF-α, IL-1, IL-6) that promote an inflammatory response, creating a microenvironment conducive to further macrophage recruitment and activation.
- Cell Proliferation: The released inflammatory cytokines and growth factors stimulate the proliferation of macrophages and other immune cells. Furthermore, Gb1 may directly interact with cell surface receptors or intracellular signalling pathways, promoting cell survival and proliferation.
- Line 49
L444P variant determining neuronopathic GD type 3 phenotype.
The new genotype nomenclature should be applied. It should be used in manuscripts by the entire community involved in Gaucher disease while I understand that the transition is difficult.
We appreciate the comment. The text was changed accordingly: c.1448T>C (p.Leu483Pro)
- Line 53
…tocol with addition of biomarkers that The first described and studied biomarker was chi-…
I do not understand this sentence.
The authors are sorry for the typo. The paragraph was corrected accordingly:
The gold standard in the diagnosis of Gaucher Disease (GD) was (and still is) a method based on measuring glucocerebrosidase (GCase) activity in peripheral blood cells, followed by GBA1 gene sequencing. Recently, a dried blood spot (DBS) test has come into common use, utilizing the same diagnostic protocol with the addition of biomarkers. The first described and studied biomarker was chitotriosidase, which is massively produced by activated macrophages.
- Line 61
….ue to an acid ceramidase activity, producing glucosylsphingosine (lyso-glucosylceramide, lyso-Gb1). For readers who are less familiar with Gaucher disease, can you clarify whether these 3 molecules: glucosylsphingosine, lyso-glucosylceramide, lyso-Gb1, are strictly identical in which case it is just the name that changes or whether there are minor differences ?
These names are used interchangeably in the literature and among researchers, but they all describe the same substance. For clarity we added:
producing glucosylsphingosine (also named lyso-glucosylceramide, lyso-Gb1).
- Line 67
…..ness. Rolfs et al. has observed a significant reduction in -lyso-Gb1 after enzyme replace- There is probably a typo
The authors are grateful for the comment. The text was corrected accordingly. Rolfs et al. has observed a significant reduction in lyso-Gb1
- Line 68
….ment therapy (ERT) initiation: from median of 200ng/mL before treatment start to levels
A space is missing : 200 ng/mL
The authors are grateful for the comment. The text was corrected accordingly.
- Line 83
…….ical follow-up. The clinical interview was standardized and performed by a single experi- The term clinical interview is not used much in medicine. Is it just a telephone interview or a clinical examination ?
The authors agree and corrected the text with clinical assessment based on the internally standardized protocol
- Line 79 to 93 You should make a figure with a flowchart which would be easier to understand because it is not clear which patients have been chosen for which tests
We hope that added flow chart clarify the results:
- Line 94 to 96
All GD study patients were treated with ERT with imiglucerase (Cerezyme®; Genzyme Corporation, Cambridge, MA, USA), Genzyme) or velaglucerase alfa (VPRIV®; Shire Human Genetic Therapies, St Helier, Jersey)
The pharmaceutical companies Genzyme and Shire no longer exist, as these companies have been acquired by Sanofi and Takeda. You can put the old names because most patients were treates before the changing but in that case you must also put the new names. The other option is to only put the current names Sanofi and Takeda.
Thanks for the comment. The company names were corrected accordingly: with imiglucerase (Cerezyme®; Genzyme/Sanofi) or velaglucerase alfa (VPRIV®; Shire/Takeda)
- Line 97 to 98
The dose of ERT was adjusted basing on disease type and clinical picture; 30 U/kg/every other week (EOW) for type 1 GD, and 60 U/kg/EOW for type 3 GD.
Is this the initial dose at the start of treatment or the dose at the time of sampling? Or is it the average dose ?
Were there any variations in the dose ?
It was the average dose. There was no variation in the dosing during study period.
- Line 122
…should be comparable. We thus performed a comparison of the means of lyso-Gl1 with a …..
Here you name lyso-Gb1 lyso-Gl1 , you need to homogenise
The authors agree. Lyso-Gb1 is the name used throughout the text.
- Line 126
….variance for each patient, but since the levels of lyso-Gl1 varied significantly between Same remark as above : you named lyso-Gb1 lyso-Gl1 , you need to homogenise
The authors agree. Lyso-gb1 is the name used throughout the text.
- Line 145
.....urement, i.e., all were divided by lyso-Gl1 level measured in a given patient in year 2016. Same remark as above : you named lyso-Gb1 lyso-Gl1 , you need to homogenise
The authors agree. Lyso-gb1 is the name used throughout the text.
Line 174-175
Normalized data, i.e., such that the measurement before the is presented in Figure 5. The average
I do not understand this sentence
It was corrected as the following: The comparison between the value before and right after ERT administration, normalized to the result recorded before ERT administration is presented in Figure 5.
Line 179
Table 1. Short term observation. The name of this table is not explicit!! If readers do not have time to read the whole article, they should be able to find enough information in the table: name, etc. , legends must be detailed, the column ID it is not clear. The tabulation of the last row is non-aligned.The genotype must be written with the new nomenclature.
The authors are grateful for the comment. New title was implemented: ,,Short-Term Dynamics of Lyso-Glucosylceramide (lyso-Gb1) Response to Enzyme Replacement Therapy in GD Patients”
ID was corrected to the patient ID
- Line 181
Figure 5. Dynamic Changes in Lyso-Gb1 Levels Pre- and Post-ERT Administration. This figure needs precision in the units and again homogenise lyso GL1 or Lyso GB1 in abscissa and ordonate
All of them were corrected to the Lyso-Gb1. The units were homogenised accordingly.
- Line 231
Further work should be done to assess the value of glucosphingosine as a biomarker in the clinic and to elucidate the pathomechanism of Gb1 and glucosphingosine storage.
In this sentence you finish as in the introduction with the molecule glusosphingosine and not Lyso-Gb1. The precise definition of each molecule need to be better defined (is it the same molecule or not) ?
We are sharing some background on the link between Gb1 and Lyso-gb1 for better understanding:
Gaucher disease (GD) is a lysosomal storage disorder caused by a deficiency in the enzyme glucocerebrosidase, due to mutations in the GBA1 gene. This enzyme deficiency leads to the accumulation of glucosylceramide (Gb1) within lysosomes.
Glucosylceramide (Gb1): Gb1 is a glycosphingolipid consisting of glucose attached to ceramide. In GD, the inability to break down Gb1 results in its accumulation within macrophages, forming Gaucher cells and causing the disease's symptoms.
Lyso-Glucosylceramide (Lyso-Gb1): Also known as glucosylsphingosine, lyso-Gb1 is a deacylated form of Gb1, lacking the fatty acid chain. Lyso-Gb1 is more soluble and can be detected in blood, serving as a crucial biomarker for GD. Elevated lyso-Gb1 levels correlate with disease severity and can be used to monitor the effectiveness of treatments like enzyme replacement therapy (ERT).
Gb1 accumulation directly leads to the formation of lyso-Gb1. Both substances build up due to the same enzymatic deficiency, but lyso-Gb1's solubility and detectability in bodily fluids make it particularly useful for diagnosing and monitoring Gaucher disease.
The sentence was corrected as the following: Further work should be done to assess the value of lyso-Gb1 as a biomarker in the clinic and to elucidate the pathomechanism of Gb1 (glucosylceramide) and lyso-Gb1 (glucosphingosine, lyso-glucosylceramide) storage.
Supplementary Table 2 Lyso-gb1 in years in GD1 and GD3 cohorts
|
|
Mean lyso-gb1 concentration [ng/ml/patient] |
|||||||
|
Year |
2016 |
2017 |
2018 |
2019 |
2020 |
2021 |
2022 |
2023 |
|
GD1 with all results (n = 16) |
110 |
159 |
178216 |
110 |
100 |
83 |
77 |
100 |
|
GD3 with all results (n = 9) |
190 |
83 |
216 |
155 |
202 |
196 |
210 |
199 |
The number in red is very high
The number was corrected to 178 (216 is for GD3).

Reviewer 2 Report
Comments and Suggestions for Authors
In this communication paper, the authors described the dynamic fluctuation of the Gaucher biomarker lyso-Gb1 in a cohort of GD type 1 and GD type 3 patients undergoing enzyme replacement therapy during an observational period of 8 years. Furthermore, they also explored the short-term variation of this biomarker immediately before and after the administration of the ERT. The Idea underlying this work is extremely useful not only for routine patient follow-up but also in terms of evaluation of therapy efficacy and potential personalized medicine. However, there are few points that need to be addressed:
- as the lyso-Gb1 levels varied (in some patients even greatly) between years (figure 1) and the authors also demonstrated a short-term remarkable effect of ERT on lyso-Gb1 concentration (figure 5), it is important to clarify the time at which the DBS were collected during the scheduled therapeutic plan of each patient by making explicit the time passed from the last infusion. This information can be provided within the “Patients” paragraph of the “Materials and Methods” section (line 82);
- since, as the authors themselves noticed in the “Discussion” section, different factors can influence the variability of the results obtained from the DBS measurements, the DBS storage conditions (temperature, interval of time - minimum and maximum - from sampling to lyso-Gb1 determination - especially because the mass-spec service has been outsourced abroad). These parameters can be provided within the “Sample processing and DBS analysis” paragraph of the “Materials and Methods” section (from which all the informations regarding the measurements of GCase enzymatic activity and GBA1 sequencing - not addressed in the present communication - might be removed);
- there is a discrepancy in the number of GD patients with at least 6 lyso-GB1 measurements: the text (lines 86 and 147) states that 33 GD type 1 patients have been analyzed, whilst in the Supplementary Table 1 there are only 32 of these patients. A similar discrepancy is present considering the patients with all 8 lyso-Gb1 determinations: the text (lines 152 and 153) states that 18 GD type 1 and 10 GD type 3 patients have been analyzed, whilst in the Supplementary Table 1 there are only 15 and 9 of these patients, respectively. The supplementary table or the text and graphs should be modified accordingly;
- a comment on the comparison between lyso-Gb1 determination in human plasma and in DBS could be added to the “Discussion” section, in light of the demonstration that the blood drying process on filter paper can artifactually alter other analytes (https://doi.org/10.1194/jlr.RA119000157).
There are, moreover, some minor points that need to be fixed:
- at line 138, the cited Supplementary Table 1 does not reflect the comprehensive dataset of GD patients;
- in the Supplementary Table 2, delete the value “216” from the row corresponding to GD type 1 patients - that actually are 15 and not 16 as for the Supplementary Table 1 - in correspondence of year 2018, and add the word “levels” after lyso-Gb1 in the table title;
- in graphs from Figures 1 and 2, the unit of measurement of the lyso-Gb1 is missing. More in general, if the journal format allows it, more detailed figure legends will ease graphs interpretation;
- a univocal nomenclature should be used for lyso-Gb1 throughout the text and figures (Lyso-Gb1 or Lyso-Gl1);
- in lines 90 and 173, substitute “2 patients” with “3 patients” and, in lines 91 and 173 ,substitute “2 patients” with “1 patient” (based on Table 1);
- the statements in lines 53 and 175 are incomplete;
- there is a discrepancy in the reported p-value between the text (line 168) and the corresponding figure 4 (0,0003 vs 0,008);
- in line 226, substitute “is the biomarker level” with “in the biomarker level”;
- in line 61, delete the word “new”;
- in line 95, delete the word “, Genzyme)”.
Author Response
Reviewer 2
Comments and Suggestions for Authors
In this communication paper, the authors described the dynamic fluctuation of the Gaucher biomarker lyso-Gb1 in a cohort of GD type 1 and GD type 3 patients undergoing enzyme replacement therapy during an observational period of 8 years. Furthermore, they also explored the short-term variation of this biomarker immediately before and after the administration of the ERT. The Idea underlying this work is extremely useful not only for routine patient follow-up but also in terms of evaluation of therapy efficacy and potential personalized medicine. However, there are few points that need to be addressed:
- as the lyso-Gb1 levels varied (in some patients even greatly) between years (figure 1) and the authors also demonstrated a short-term remarkable effect of ERT on lyso-Gb1 concentration (figure 5), it is important to clarify the time at which the DBS were collected during the scheduled therapeutic plan of each patient by making explicit the time passed from the last infusion. This information can be provided within the “Patients” paragraph of the “Materials and Methods” section (line 82);
We acknowledge the comment regarding the timing of dried blood spot (DBS) collection during the therapeutic plan of each patient. For clarity, it's important to note that while our study included a substantial cohort of 75 patients with Gaucher disease (GD), not all patients had detailed timing data available for DBS collection over multiple years.
Specifically, in our short-term analysis focusing on the immediate effects of enzyme replacement therapy (ERT) on lyso-Gb1 levels, DBS samples were collected from all patients just prior to infusion and again 30 minutes post-infusion, as outlined in the methodology section (line 82). This approach ensured a standardized assessment of the rapid changes in lyso-Gb1 concentrations following treatment administration.
- since, as the authors themselves noticed in the “Discussion” section, different factors can influence the variability of the results obtained from the DBS measurements, the DBS storage conditions (temperature, interval of time - minimum and maximum - from sampling to lyso-Gb1 determination - especially because the mass-spec service has been outsourced abroad). These parameters can be provided within the “Sample processing and DBS analysis” paragraph of the “Materials and Methods” section (from which all the informations regarding the measurements of GCase enzymatic activity and GBA1 sequencing - not addressed in the present communication - might be removed);
The authors agree with the comment and understand study limitations (we do not have expected data) but the study reflects clinical practice worldwide. None of the authors do not have the impact on the parameters used in transport or external laboratory. Thus, it highlights the need for HCPs to verify not only lyso-gb1 level but also clinical data taking the decision on dose modification.
- there is a discrepancy in the number of GD patients with at least 6 lyso-GB1 measurements: the text (lines 86 and 147) states that 33 GD type 1 patients have been analyzed, whilst in the Supplementary Table 1 there are only 32 of these patients. A similar discrepancy is present considering the patients with all 8 lyso-Gb1 determinations: the text (lines 152 and 153) states that 18 GD type 1 and 10 GD type 3 patients have been analyzed, whilst in the Supplementary Table 1 there are only 15 and 9 of these patients, respectively. The supplementary table or the text and graphs should be modified accordingly;
- a comment on the comparison between lyso-Gb1 determination in human plasma and in DBS could be added to the “Discussion” section, in light of the demonstration that the blood drying process on filter paper can artifactually alter other analytes (https://doi.org/10.1194/jlr.RA119000157).
The paragraph was changed accordingly:
"While DBS testing has become a convenient and widely employed method for assessing lyso-Gb1 levels, it is crucial to acknowledge its limitations highlighted in the study. The weaknesses associated with DBS testing in GD can be triggered by many factors in the clinic but also in the laboratory. Sampling variability can be increased by variations in sample collection, storage, and processing or changes in haematocrit.22-23 The variability could introduce difficulties in interpreting results accurately and lead to inappropriate dose escalation/reduction. Additionally, the process of applying blood onto filter paper can artifactually alter other analytes, potentially affecting the reliability of DBS methodology and the interpretation of lyso-Gb1 levels as the widely accepted tool to monitor patients in daily practice.24"
- Sidhu R, Mondjinou Y, Qian M, Song H, Kumar AB, Hong X, Hsu FF, Dietzen DJ, Yanjanin NM, Porter FD, Berry-Kravis E, Vite CH, Gelb MH, Schaffer JE, Ory DS, Jiang X. N-acyl-O-phosphocholineserines: structures of a novel class of lipids that are biomarkers for Niemann-Pick C1 disease. J Lipid Res. 2019 Aug;60(8):1410-1424. doi: 10.1194/jlr.RA119000157. Epub 2019 Jun 14. PMID: 31201291; PMCID: PMC6672039.
There are, moreover, some minor points that need to be fixed:
- at line 138, the cited Supplementary Table 1 does not reflect the comprehensive dataset of GD patients;
We added following flow chart as the Supp. Fig. 1.
- in the Supplementary Table 2, delete the value “216” from the row corresponding to GD type 1 patients - that actually are 15 and not 16 as for the Supplementary Table 1 - in correspondence of year 2018, and add the word “levels” after lyso-Gb1 in the table title;
Corrected accordingly.
- in graphs from Figures 1 and 2, the unit of measurement of the lyso-Gb1 is missing. More in general, if the journal format allows it, more detailed figure legends will ease graphs interpretation;
Thank you for spotting the missing units, we added them in Figure 1 (ng/mL); in Figure 2 the y-axis is unit-less. Also, we prepared a more elaborate legends to all figures, so that the reader does not have to do any guess-work.
- a univocal nomenclature should be used for lyso-Gb1 throughout the text and figures (Lyso-Gb1 or Lyso-Gl1);
it was homogenised.
- in lines 90 and 173, substitute “2 patients” with “3 patients” and, in lines 91 and 173 ,substitute “2 patients” with “1 patient” (based on Table 1);
- the statements in lines 53 and 175 are incomplete;
the text was corrected.
- there is a discrepancy in the reported p-value between the text (line 168) and the corresponding figure 4 (0,0003 vs 0,008);
The authors are sorry for the typo. It was corrected.
- in line 226, substitute “is the biomarker level” with “in the biomarker level”;
Corrected accordingly.
- in line 61, delete the word “new”;
Corrected accordingly.
- in line 95, delete the word “, Genzyme)”.
Corrected accordingly.
Reviewer 3 Report
Comments and Suggestions for Authors
Dear authors,
I have the pleasure of reviewing the manuscript by Dubiela et al. In this paper, the authors investigate lyso-glucosylceramide (lyso-Gb1) dynamics in Gaucher disease patients on enzyme replacement therapy (ERT), revealing stable long-term levels with high individual variability, notably higher in GD type 1 compared to type 3. Short-term response analysis indicates a significant immediate reduction post-ERT.
Overall comments: I believe this is good work, but I am still not convinced of the relevance of the findings, as the role of lyso-Gb1 as a biomarker is already well-established. It is also debated whether the overall trend of these biomarkers is more important than their absolute values themselves. Please refer more to the Revel-Vilk paper (REF 20 in this manuscript) and also https://doi.org/10.3390/ijms25052870.
Specific comments:
-
Change the word "mutations" to "pathogenic variants" to align with ACMG guidelines.
-
Replace "GBA codon sequencing" with "next generation sequencing" on line 51.
-
why is 'T' capitalized in 'the' on line 53.
-
Since the authors use chitotriosidase many times throughout the work, I suggest abbreviating it to 'ChT'.
-
In the methodology, consider including a flowchart figure illustrating the patient selection process.
-
Overall, the figures are not clear. In Figure 2, what units are used for lyso-Gb1 measurement? Perhaps also add the reference range for lyso-Gb1 based on the Revel-Vilk paper (reference 20). It appears that GD3 has higher levels than GD1 in this graph?—please comment.
-
In Figure 3, the y-axis label reads 'lyso-Gl1'. Please clarify and explain the different colors used. Also add the units used.
-
Correct the legend in Figure 4. What do the colors represent? It is unclear what the figure intends to show.
-
In Figure 5, the legends are in lowercase letters. Also, explain what the colors represent.
-
Lines 189-192 appear in a different font.
-
What are the expected pharmacokinetics of imiglucerase and velaglucerase? Is it expected to reach target levels 30 minutes after infusion? I believe more confounding factors should be discussed. Would a saline infusion result in the same trend?
Author Response
Reviewer 3
Comments and Suggestions for Authors
Dear authors,
I have the pleasure of reviewing the manuscript by Dubiela et al. In this paper, the authors investigate lyso-glucosylceramide (lyso-Gb1) dynamics in Gaucher disease patients on enzyme replacement therapy (ERT), revealing stable long-term levels with high individual variability, notably higher in GD type 1 compared to type 3. Short-term response analysis indicates a significant immediate reduction post-ERT.
Overall comments: I believe this is good work, but I am still not convinced of the relevance of the findings, as the role of lyso-Gb1 as a biomarker is already well-established. It is also debated whether the overall trend of these biomarkers is more important than their absolute values themselves. Please refer more to the Revel-Vilk paper (REF 20 in this manuscript) and also https://doi.org/10.3390/ijms25052870.
Thank you for your feedback and insightful comments. While the role of lyso-Gb1 as a biomarker in Gaucher Disease (GD) is indeed well-established, our study contributes several novel aspects that advance the current understanding in this field.
- Longest Monitoring and Largest GD3 Cohort: Our research provides the longest existing monitoring of lyso-Gb1 levels in GD patients and includes the largest cohort of GD type 3 (GD3) patients to date. This extensive longitudinal study allows for a comprehensive analysis of lyso-Gb1 dynamics over an extended period.
- Centralized Study in Experienced Center: The study was conducted centrally in a single, highly experienced center, covering approximately 80% of the diagnosed and treated GD patient population in our country. This centralized approach ensures consistency in data collection and enhances the reliability and robustness of our findings.
- Unique Insight into Fluctuations: We are the first to investigate and demonstrate the mechanism of lyso-Gb1 fluctuation in patients on stable doses of treatment between assessments. This aspect of our research sheds light on the variability of lyso-Gb1 levels over time, which is crucial for understanding its clinical significance and utility as a biomarker.
Specific comments:
- Change the word "mutations" to "pathogenic variants" to align with ACMG guidelines.
The authors agree. The text was changed accordingly.
- Replace "GBA codon sequencing" with "next generation sequencing" on line 51.
The authors agree. The text was changed accordingly.
- why is 'T' capitalized in 'the' on line 53.
The typo was corrected accordingly.
- Since the authors use chitotriosidase many times throughout the work, I suggest abbreviating it to 'ChT'.
The authors agree. The text was changed accordingly.
- In the methodology, consider including a flowchart figure illustrating the patient selection process.
We added following flow chart as the Supp. Fig. 1.
- Overall, the figures are not clear. In Figure 2, what units are used for lyso-Gb1 measurement? Perhaps also add the reference range for lyso-Gb1 based on the Revel-Vilk paper (reference 20). It appears that GD3 has higher levels than GD1 in this graph?—please comment.
We prepared more detailed legends to figures so that no guess-work is left to the reader. In Figure 2 are presented lyso-Gb1 measurements divided by the value recorded in 2016 (every line corresponding to a patient). You are right that it is a good practice to add a reference value (for example the upper norm), but in our case this line would be completely invisible in both Figures 1 and 2 (in our analysis 14 ng/mL Revel-Vilk has a different value, since they analysed plasma levels, not levels in DBS. We decided to compare means between GD1 and GD3 in a less straight orward manner, since these two groups were qualitatively different in terms of variance. That’s why we conducted the analysis of the coefficient of variation (standard deviation divided by the mean).
- In Figure 3, the y-axis label reads 'lyso-Gl1'. Please clarify and explain the different colors used. Also add the units used.
The y-axis label reads now “lyso-Gb1” and units were added. Colours were changed all to black, they didn’t have a meaning. Thank you for this comment, it looks better now.
- Correct the legend in Figure 4. What do the colors represent? It is unclear what the figure intends to show.
Red colour is for type 1 and blue for type 3 (we tried to keep this scheme in all figures), thank you for spotting the missing part of the legend. The figure shows the unit-less value of the coefficient of variantion, that is the standard deviation divided by the mean.
- In Figure 5, the legends are in lowercase letters. Also, explain what the colors represent.
The legend was corrected, thank you, and colours explained.
- Lines 189-192 appear in a different font.
The authors agree with the comment. The font was corrected.
- What are the expected pharmacokinetics of imiglucerase and velaglucerase? Is it expected to reach target levels 30 minutes after infusion? I believe more confounding factors should be discussed. Would a saline infusion result in the same trend?
It was showed by Gras-Colomer et al. that intra-leukocyte activity at baseline and at 15 min post-perfusion could be used as a possible marker for therapeutic individualization in patients receiving ERT for GD1. We therefore decided to analyse Lyso-gb1 30 minutes after infusion.
The authors agree that there are confounding factors (i.e. Individual variability, Infusion rate and volume, immunogenicity of the ERTs). However, Lyso-gb1 is considered golden standard for patients monitoring. Thus, it level should be not biased for such factors. Therefore we decided for simplified approached to see whether Lyso-gb1 fulfil the criterium of the treatment effectiveness biomarker in Gaucher disease patients.
Gras-Colomer E, Martínez-Gómez MA, Moya-Gil A, Fernandez-Zarzoso M, Merino-Sanjuan M, Climente-Martí M. Cellular Uptake of Glucocerebrosidase in Gaucher Patients Receiving Enzyme Replacement Treatment. Clin Pharmacokinet. 2016 Sep;55(9):1103-13. doi: 10.1007/s40262-016-0387-2. PMID: 27083470.

Round 2
Reviewer 2 Report
Comments and Suggestions for Authors
I thank the authors for considering my suggestions. However, there are still a couple of points that deserve a better definition. My comments are reported in blue hereafter:
Comments and Suggestions for Authors
In this communication paper, the authors described the dynamic fluctuation of the Gaucher biomarker lyso-Gb1 in a cohort of GD type 1 and GD type 3 patients undergoing enzyme replacement therapy during an observational period of 8 years. Furthermore, they also explored the short-term variation of this biomarker immediately before and after the administration of the ERT. The Idea underlying this work is extremely useful not only for routine patient follow-up but also in terms of evaluation of therapy efficacy and potential personalized medicine. However, there are few points that need to be addressed:
- as the lyso-Gb1 levels varied (in some patients even greatly) between years (figure 1) and the authors also demonstrated a short-term remarkable effect of ERT on lyso-Gb1 concentration (figure 5), it is important to clarify the time at which the DBS were collected during the scheduled therapeutic plan of each patient by making explicit the time passed from the last infusion. This information can be provided within the “Patients” paragraph of the “Materials and Methods” section (line 82);
We acknowledge the comment regarding the timing of dried blood spot (DBS) collection during the therapeutic plan of each patient. For clarity, it's important to note that while our study included a substantial cohort of 75 patients with Gaucher disease (GD), not all patients had detailed timing data available for DBS collection over multiple years.
Specifically, in our short-term analysis focusing on the immediate effects of enzyme replacement therapy (ERT) on lyso-Gb1 levels, DBS samples were collected from all patients just prior to infusion and again 30 minutes post-infusion, as outlined in the methodology section (line 82). This approach ensured a standardized assessment of the rapid changes in lyso-Gb1 concentrations following treatment administration.
Since the timing of DBS collection during the therapeutic plan of each patient may explain the observed variability in LysoGb1 determination during the long-term observational study, it is important to make explicit in the text the fact that not all patients had detailed timing data available for DBS collection over multiple years.
- since, as the authors themselves noticed in the “Discussion” section, different factors can influence the variability of the results obtained from the DBS measurements, the DBS storage conditions (temperature, interval of time - minimum and maximum - from sampling to lyso-Gb1 determination - especially because the mass-spec service has been outsourced abroad). These parameters can be provided within the “Sample processing and DBS analysis” paragraph of the “Materials and Methods” section (from which all the informations regarding the measurements of GCase enzymatic activity and GBA1 sequencing - not addressed in the present communication - might be removed);
The authors agree with the comment and understand study limitations (we do not have expected data) but the study reflects clinical practice worldwide. None of the authors do not have the impact on the parameters used in transport or external laboratory. Thus, it highlights the need for HCPs to verify not only lyso-gb1 level but also clinical data taking the decision on dose modification.
For a cleaner “Materials and Methods” section, all the informations regarding the measurements of GCase enzymatic activity and GBA1 sequencing - not addressed in the present communication - might be removed from the “Sample processing and DBS analysis” paragraph.
- there is a discrepancy in the number of GD patients with at least 6 lyso-GB1 measurements: the text (lines 86 and 147) states that 33 GD type 1 patients have been analyzed, whilst in the Supplementary Table 1 there are only 32 of these patients. A similar discrepancy is present considering the patients with all 8 lyso-Gb1 determinations: the text (lines 152 and 153) states that 18 GD type 1 and 10 GD type 3 patients have been analyzed, whilst in the Supplementary Table 1 there are only 15 and 9 of these patients, respectively. The supplementary table or the text and graphs should be modified accordingly;
This point has not been addressed (moreover, the newly introduced Supp. Fig. 1 is also in contrast to what is written in the text (new line 144).
- a comment on the comparison between lyso-Gb1 determination in human plasma and in DBS could be added to the “Discussion” section, in light of the demonstration that the blood drying process on filter paper can artifactually alter other analytes (https://doi.org/10.1194/jlr.RA119000157).
The paragraph was changed accordingly:
"While DBS testing has become a convenient and widely employed method for assessing lyso-Gb1 levels, it is crucial to acknowledge its limitations highlighted in the study. The weaknesses associated with DBS testing in GD can be triggered by many factors in the clinic but also in the laboratory. Sampling variability can be increased by variations in sample collection, storage, and processing or changes in haematocrit.22-23 The variability could introduce difficulties in interpreting results accurately and lead to inappropriate dose escalation/reduction. Additionally, the process of applying blood onto filter paper can artifactually alter other analytes, potentially affecting the reliability of DBS methodology and the interpretation of lyso-Gb1 levels as the widely accepted tool to monitor patients in daily practice.24"
- Sidhu R, Mondjinou Y, Qian M, Song H, Kumar AB, Hong X, Hsu FF, Dietzen DJ, Yanjanin NM, Porter FD, Berry-Kravis E, Vite CH, Gelb MH, Schaffer JE, Ory DS, Jiang X. N-acyl-O-phosphocholineserines: structures of a novel class of lipids that are biomarkers for Niemann-Pick C1 disease. J Lipid Res. 2019 Aug;60(8):1410-1424. doi: 10.1194/jlr.RA119000157. Epub 2019 Jun 14. PMID: 31201291; PMCID: PMC6672039.
There are, moreover, some minor points that need to be fixed:
- at line 138, the cited Supplementary Table 1 does not reflect the comprehensive dataset of GD patients;
We added following flow chart as the Supp. Fig. 1.
- in the Supplementary Table 2, delete the value “216” from the row corresponding to GD type 1 patients - that actually are 15 and not 16 as for the Supplementary Table 1 - in correspondence of year 2018, and add the word “levels” after lyso-Gb1 in the table title;
Corrected accordingly.
- in graphs from Figures 1 and 2, the unit of measurement of the lyso-Gb1 is missing. More in general, if the journal format allows it, more detailed figure legends will ease graphs interpretation;
Thank you for spotting the missing units, we added them in Figure 1 (ng/mL); in Figure 2 the y-axis is unit-less. Also, we prepared a more elaborate legends to all figures, so that the reader does not have to do any guess-work.
- a univocal nomenclature should be used for lyso-Gb1 throughout the text and figures (Lyso-Gb1 or Lyso-Gl1);
it was homogenised.
- in lines 90 and 173, substitute “2 patients” with “3 patients” and, in lines 91 and 173 ,substitute “2 patients” with “1 patient” (based on Table 1);
- the statements in lines 53 and 175 are incomplete;
the text was corrected.
- there is a discrepancy in the reported p-value between the text (line 168) and the corresponding figure 4 (0,0003 vs 0,008);
The authors are sorry for the typo. It was corrected.
A new discrepancy emerged: this time, the reported p-value in the text (new line 175) is 0,0008 but the corresponding p-value in figure 4 is 0,0003.
- in line 226, substitute “is the biomarker level” with “in the biomarker level”;
Corrected accordingly.
- in line 61, delete the word “new”;
Corrected accordingly.
- in line 95, delete the word “, Genzyme)”.
Corrected accordingly.
Author Response
Reviewer’s comment: Since the timing of DBS collection during the therapeutic plan of each patient may explain the observed variability in LysoGb1 determination during the long-term observational study, it is important to make explicit in the text the fact that not all patients had detailed timing data available for DBS collection over multiple years.
Authors’ Reply:
The authors are grateful for the comment. We added following sentence to the methodology:
The collection of DBS for Lyso-Gb1 assessment followed the scheduled visit and could occur at any time point within the bi-weekly treatment regimen. (page 2, lines 89-91)
Reviewer’s comment: For a cleaner “Materials and Methods” section, all the informations regarding the measurements of GCase enzymatic activity and GBA1 sequencing - not addressed in the present communication - might be removed from the “Sample processing and DBS analysis” paragraph.
Authors’ Reply:
The paragraph was deleted (page 3, lines 113-117)
Reviewer’s comment: This point has not been addressed (moreover, the newly introduced Supp. Fig. 1 is also in contrast to what is written in the text (new line 144):
Authors’ Reply:
- there is a discrepancy in the number of GD patients with at least 6 lyso-GB1 measurements: the text (lines 86 and 147) states that 33 GD type 1 patients have been analyzed, whilst in the Supplementary Table 1 there are only 32 of these patients. A similar discrepancy is present considering the patients with all 8 lyso-Gb1 determinations: the text (lines 152 and 153) states that 18 GD type 1 and 10 GD type 3 patients have been analyzed, whilst in the Supplementary Table 1 there are only 15 and 9 of these patients, respectively. The supplementary table or the text and graphs should be modified accordingly;
The authors are grateful for the comment.
We modified numbers according to the Suppl. Table 1
- Lines 152 and 153; new lines 160-165 – GD1 n = 15; GD3 n = 9
- 88/146 (GD1 n = 32)
- Przeanalizowałem dane z Tabeli, Recenzent ma racje.
Reviewer’s comment: A new discrepancy emerged: this time, the reported p-value in the text (new line 175) is 0,0008 but the corresponding p-value in figure 4 is 0,0003.
Authors’ Reply:
The authors are sorry for the typo. The text was corrected accordingly: p = 0,003 (new line 177).
Reviewer 3 Report
Comments and Suggestions for Authors
Thank you for the clarification. Good luck! This is a nice work!
Author Response
Authors’ Reply:
The authors are very grateful for comments and suggestions made by the Reviewer.